# Molecular and Cellular Characterization of Avian Reticulate Scales Implies the Evo–Devo Novelty of Skin Appendages in Foot Sole

**DOI:** 10.3390/jdb11030030

**Published:** 2023-07-03

**Authors:** Tzu-Yu Liu, Michael W. Hughes, Hao-Ven Wang, Wei-Cheng Yang, Cheng-Ming Chuong, Ping Wu

**Affiliations:** 1Department of Life Sciences, National Cheng Kung University, Tainan 701, Taiwan; brothansis@yahoo.com.tw (T.-Y.L.); hvwang@ncku.edu.tw (H.-V.W.); 2Marine Biology and Cetacean Research Center, National Cheng Kung University, Tainan 701, Taiwan; 3Institute of Clinical Medicine and Department of Life Sciences, College of Medicine, National Cheng Kung University, Tainan 701, Taiwan; mwhughes@usc.edu; 4School of Veterinary Medicine, National Taiwan University, Taipei 106216, Taiwan; yangweicheng@ntu.edu.tw; 5Department of Pathology, Keck School of Medicine, University Southern California, Los Angeles, CA 90033, USA

**Keywords:** scale, bird, reptile, stem cell

## Abstract

Among amniotic skin appendages, avian feathers and mammalian hairs protect their stem cells in specialized niches, located in the collar bulge and hair bulge, respectively. In chickens and alligators, label retaining cells (LRCs), which are putative stem cells, are distributed in the hinge regions of both avian scutate scales and reptilian overlapping scales. These LRCs take part in scale regeneration. However, it is unknown whether other types of scales, for example, symmetrically shaped reticulate scales, have a similar way of preserving their stem cells. In particular, the foot sole represents a special interface between animal feet and external environments, with heavy mechanical loading. This is different from scutate-scale-covered metatarsal feet that function as protection. Avian reticulate scales on foot soles display specialized characteristics in development. They do not have a placode stage and lack β-keratin expression. Here, we explore the molecular and cellular characteristics of avian reticulate scales. RNAscope analysis reveals different molecular profiles during surface and hinge determination compared with scutate scales. Furthermore, reticulate scales express Keratin 15 (*K15*) sporadically in both surface- and hinge-region basal layer cells, and LRCs are not localized. Upon wounding, the reticulate scale region undergoes repair but does not regenerate. Our results suggest that successful skin appendage regeneration requires localized stem cell niches to guide regeneration.

## 1. Introduction

Amniote skins are multi-layered, and cells are continuously shed from the skin surface. Amniotes present different types of skin appendages, including scales, feathers, and hairs [1]. Reptile scales are considered the basal skin appendage from which avian feathers and mammalian hairs evolved (Figure 1A) [1,2]. Birds exhibit scales on their feet, which include two main types. The scutate scales in the metatarsal region are overlapping and resemble the overlapping scales in reptiles. Their main function is protection. The reticulate scales on the foot sole are dome-shaped, resembling reptilian tuberculate scales. Their main function is weight bearing and dealing with constant friction and minor injuries. It is interesting to compare the topology and homeostasis of stem/transit-amplifying/differentiated cells in different skin appendages and to appreciate the Evo–Devo of different skin appendages with different functions. 

Feathers and hair share many common characteristics, although they evolved independently from reptilian scales. Some of these characteristics include periodic molting, the topology of their stem cells and transit-amplifying cells (TA cells), and the presence of the dermal papilla [3,4]. Avian feathers and mammalian hair maintain stem cells in a supportive niche within their follicular structure [4,5,6]. The stem cell compartment is localized within the bulge of hair follicles [7] and within the collar bulge of feather follicles [6].

**Figure 1 jdb-11-00030-f001:**
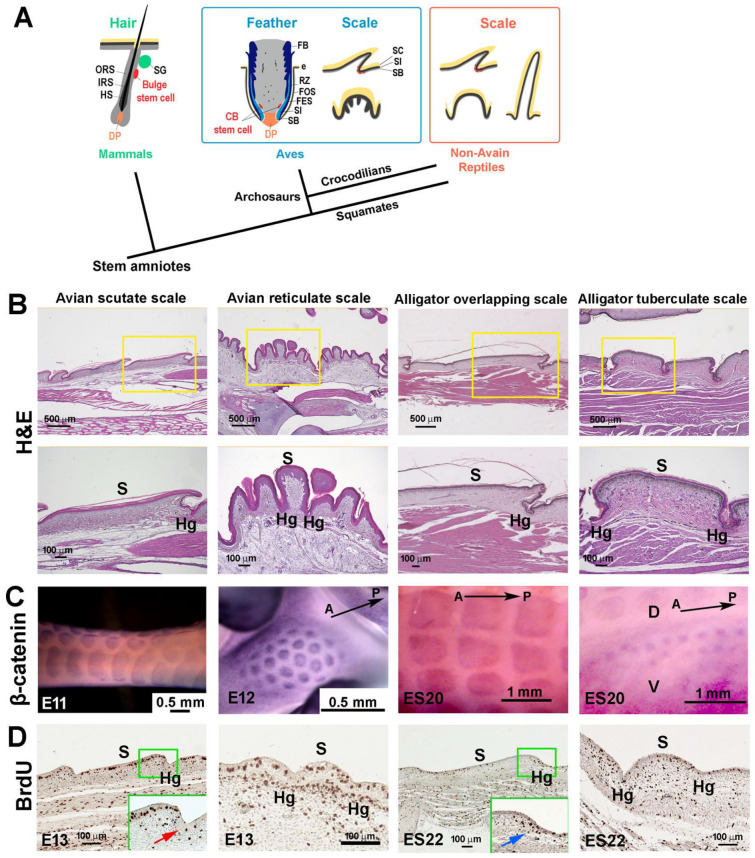
Development of amniote scales: avian scutate scales, avian reticulate scales, reptilian overlapping scales, and reptilian tuberculate scales. (**A**) Schematic drawing of the architecture of stem cells and niches in amniote skin appendages (modified from [8]). (**B**) H&E staining showing the chicken scutate scale, reticulate scale, alligator overlapping scale, and alligator tuberculate scale in hatchling chickens and alligators. (**C**) β-catenin whole-mount in situ hybridization showing the scale primordia. (**D**) BrdU staining showing the distribution of proliferation cells in developing scales. Note the hinge region has fewer proliferation cells in chicken scutate scales (indicated by the red arrow) whereas the hinge and outer surface regions have similar cell proliferation patterns in chicken reticulate scales and alligator tuberculate scales. There are condensed BrdU-positive cells in the hinge dermal cells of ES22 alligator developing overlapping scales (blue arrow). CB, collar bulge; DP, dermal papilla; e, epidermis; FB; feather barb ridge; FES, feather sheath; FOS, feather follicle sheath; HS, hair shaft; IRS, inner root sheath; M, dorsal middle line of alligator embryo; ORS, outer root sheath; RZ, ramogenic zone; SG, sebaceous gland; SB, stratum basal; SC, stratum corneum; SI, stratum intermedium. A, anterior; D, dorsal; Hg, hinge; P, posterior; S, surface; V, ventral.

Previously, we examined avian scutate scales and reptilian overlapping scales and concluded that both scale types display diffuse patterns of molecular expression and cell proliferation compared with those of avian feathers. Using a BrdU label retention method to locate slow-cycling cells, we identified a similarly diffuse putative stem cell niche in morphologically similar chicken scutate and alligator overlapping scales. These putative stem cells participate in alligator scale regeneration [8].

However, we still lack understanding of one mysterious skin appendage, the avian reticulate scale. This scale type displays special characteristics during development. For example, reticulate scales have epidermal ridges, do not form thickened epidermal placodes in early skin appendage primordia, and do not express β-keratins [9]. The development of squamate footpad scales and avian reticulate scales are similarly derived epithelial appendages [10]. We found that avian reticulate scales do not express β-keratins except for keratinocyte-β-keratin 13, which is expressed in both the scale hinge and surface [11]. 

Chicken scales can be converted into skin appendages with a feather phenotype. Several molecular pathway perturbations, including retinoic acid, Wnt/β-catenin, Notch/Delta pathway activation, BMP pathway suppression, and Shh pathway activation, can induce feather formation from scutate scales [12,13,14,15,16]. We also showed a set of novel scale–feather converters (*Sox2*, *Zic1*, *Grem1*, *Spry2*, *Sox18*) which induce feather-like skin appendages to form from scales to different extents [17]. Only a few feather-to-scale converters, for example, retinoic acid or Shh agonist, can induce feathers to form from reticulate scales [12,16]. These studies exhibit the molecular differences between scutate and reticulate scales. We hypothesize that specific molecular and cellular characteristics of reticulate scales in footpads, formed in the developing skin, set up the bio-architectural basis for adult birds to bear their weight and to withstand constant friction.

In this paper, our objective is to assess the molecular expression and putative stem cell configurations of avian reticulate scales in comparison to other chicken and alligator scale types. We find that reticulate scales have different LRC properties and regeneration abilities compared with these other scale types.

## 2. Method 

### 2.1. Juvenile Alligators and Adult Chickens

We used adult chickens and juvenile alligators (6 months to 1 year old) for TA and LRC labeling. Alligator eggs were collected from the Rockefeller Wildlife Refuge in Louisiana. Eggs were transported to USC and incubated at 30 °C. All procedures were approved by the local Institutional Animal Care and Use Committee at the University of Southern California. For chicken experiments, the IACUC protocol number is 20231 and the approval date is 23 July 2020. For alligator experiment, the IACUC protocol number is 10736 and the approval date is 29 November 2022.

### 2.2. Pulse BrdU Labeling and Identification of Label Retaining Cells

For BrdU staining of chicken and alligator embryos, 10 µL 1% BrdU was injected into a vein. After 2 h, the embryos were fixed in 4% paraformaldehyde and prepared for sectioning.

For the pulse labeling of juvenile alligators and adult chickens, BrdU was injected intraperitoneally at 50 mg per kg (body weight). Scales were collected 3 h later. For the label-retention studies, animals were injected with BrdU per day for 1 week and ‘chased’ (allowed to metabolize the BrdU in their system) for up to 8 weeks for chickens and 16 weeks for alligators. One chicken and one alligator were euthanized after one week of BrdU labeling (one-week pulse). Four adult chickens and four juvenile alligators were used for the LRC study. BrdU was detected by immunostaining [18].

### 2.3. Immunostaining and Whole-Mount/Section In Situ Hybridization

For immunostaining, section in situ hybridization, and RNAscope experiments, 7 µm paraffin sections were prepared. Immunostaining of Tenascin-C (*TNC*) was performed according to [19]. A β-catenin (*CTNNB1*) RNA probe from chicken [13] and alligator [19] was used for whole-mount in situ hybridization. Chicken and alligator Keratin 15 (*K15*) and Keratin 75 (*K75*) probes were from [8]. Whole-mount and section in situ hybridization were performed according to described procedures [20]. Diluted eosin was used as a faint counter-staining. 

### 2.4. RNAscope

RNAscope was performed using the Multiplex Fluorescent v2 system (323100, Advanced Cell Diagnostics, Newark, NJ, USA). The standard RNAscope protocol was used according to the manufacturer’s instructions. We used the following probes: *LGR4* (1097771-C2), *LGR5* (480781-C1), *LGR6* (1097781-C3), *NOG* (480101-C1), *SOSTDC1* (1055361-C2), *TGFβ2* (1055431-C3). Confocal images were generated with a Leica TCS SP8 confocal microscope (Leica Microsystems, Morrisville, NC, USA).

### 2.5. Transgenic Quail Eggs and Confocal Imaging

Fertilized transgenic quail eggs with MEM-GFP (membrane-bound EGFP under the control of the ubiquitous human ubiquitin C promoter, green color) [21] and H2B-chFP (ubiquitously expresses nuclear-localized monomer Cherry fluorescent protein, red color) [22] were provided by Dr. Rusty Lansford at USC. E11 embryos were collected, and the fluorescent signals were observed with a Leica TCS SP8 confocal microscope. 

### 2.6. Reticulate Scale Wound Healing and Regeneration

For reticulate scale wounding and regeneration, chickens were anesthetized by an intramuscular injection of ketamine (50 mg/kg) and xylazine (5 mg/kg). Biopsies (1 cm wide and 1 cm long) were traced with a scalpel to about 1 mm in depth. The skin was lifted with forceps and excised with a scalpel. Animals were euthanized after five months of regeneration. Paraffin sections were prepared for H&E staining and immunostaining. Three adult chickens were used.

## 3. Results

### 3.1. The Structure of Reticulate Scales and Comparison with Other Scales

Modern birds have two main kinds of scales on their feet: scutate scales on the tarsometatarsal region and reticulate scales on the footpad (Figure 1B, left and second column). Compared with overlapping-shaped scutate scales, reticulate scales have a symmetrical shape and are smaller in size, with epidermal ridges inserting into the scale dermis. The tuberculate scales on the lateral side of the alligator’s body show a similar symmetrical shape but do not have epidermal ridges (Figure 1B, right column), whereas overlapping scales (Figure 1B, third column) in the dorsal region have an asymmetrical shape like avian scutate scales. β-catenin whole-mount in situ hybridization showed that the expression pattern is more rectangular-shaped in chicken scutate scale and alligator overlapping scale primordia, and round-shaped in both chicken reticulate and alligator tuberculate scale primordia (Figure 1C). Transit-amplifying cells (TA cells) labeled by short-term BrdU showed a more even distribution of proliferating cells in both chicken reticulate and alligator tuberculate scale primordia, unlike the surface vs. hinge differential distribution in scutate scales (red arrow) and alligator overlapping scales (blue arrow) (Figure 1D).

### 3.2. Difference in Morphogen Expression between Avian Scutate Scales and Reticulate Scales

To examine the expression of morphogens in chicken reticulate scales, we performed RNAscope analysis. The six candidate molecules are involved in skin stem cell regulation and skin appendage morphogenesis [23,24,25,26]. We compared the development of avian scutate scales with that of reticulate scales in the early (E11), middle (E14), and late (E17) developmental stages. 

The Leucine-Rich Repeat-Containing G-Protein-Coupled Receptors (LGRs) are receptors for R-spondins that function through the canonical Wnt signaling pathway. They have been found to be adult stem cell markers in several cell types [23]. The analysis of the first set of probes (*LGR4*, *LGR5*, *LGR6*) showed that LGR4 and LGR6 were expressed at high levels in the early- and middle-stage scutate scales (Figure 2A,B) and in middle- and late-stage reticulate scales (Figure 2E,F). In contrast, LGR5 did not show a specific expression pattern in developing scutate or reticulate scales. We found differential expression of LGR6 on the surface of the scutate scales at the early and middle stages (Figure 2A,B, red arrows). In contrast, LGR6 is expressed on the surface of reticulate scales in the middle stage (Figure 2E, blue arrow) and then expands to the entire reticulate scale epidermis in the late stage (Figure 2F, white arrow). 

The second set of probes included *NOG*, *SOSTDC1*, and *TGFβ2*. Noggin (*NOG*) is an inhibitor of bone morphogenetic protein (BMP) that has been shown to be involved in avian feather branching formation [24]. Sclerostin Domain-Containing 1 (*SOSTDC1*) is a BMP and Wnt pathway modulator and controls the size and number of skin appendage placodes in mice [26]. Transforming growth factor-beta 2 (*TGFβ2*) plays an important role in the induction of dermal condensation in embryonic feather development [25]. NOG showed faint expression in the epidermis, while TGFβ2 showed a dermal expression pattern in both chicken scutate and reticulate scales. No clear differences between scutate and reticulate scales were detected for NOG and TGFβ2 expression. However, we found that SOSTDC1 is differentially expressed in the hinge area of the scutate scales at the early and middle stages (yellow arrows, Figure 2G,H) and then expressed in the entire epithelium at the late stage, but is present throughout the reticulate scale epidermis at early, middle, and late stages (green arrows, Figure 2J–L). These results suggest that the epidermis of reticulate scales lacks the surface and hinge difference seen in scutate scales. 

### 3.3. Difference in Epidermal Cell Arrangements in Developing Scutate and Reticulate Scales Using Transgenic Quail Embryos 

We used transgenic quail embryos which express MEM-GFP (membrane-bound EGFP under the control of the ubiquitous human ubiquitin C promoter, green color) and H2B-chFP (ubiquitously expresses nuclear-localized monomer Cherry fluorescent protein, red color) to observe the epidermal cell shape (Figure 3). The epidermal cells had regular cell shapes and tissue patterning in scutate scales (Figure 3A,B). The cell shapes and tissue patterns were irregular in reticulate scales (Figure 3C,D). These data imply differences in regional-specific tissue patterning due to epidermal cell behavior in the development of scutate scales versus reticulate scales.

### 3.4. The Expression of K15 Is Sporadic and Not Restricted in the Reticulate Scale Hinge like in the Scutate Scale

The intermediate filament keratin 15 (K15) has been used as a hair follicle stem cell marker [27,28]. Previously, we found K15 to have a restricted expression pattern in the hinge of chicken scutate scales and alligator overlapping scales. Additionally, another alpha keratin, Keratin 75 (*K75*), was found to be expressed in the more differentiated supra-basal layer of the outer surface [8].

In reticulate scales, we found that K15 is sporadically expressed in the basal layer cells in both the hinge (red arrows) and surface (green arrows) regions (Figure 4A, upper panels). K75 is expressed in the supra-basal layer in both the hinge and surface (Figure 4A, lower panels).

We further examined the expression patterns of K15 and K75 in alligator tuberculate scales (Figure 4B). These expression patterns are similar to the chicken scutate scales and alligator overlapping scales. K15 is only expressed in the hinge basal layer cells, not in the surface basal layer cells, and K75 is expressed in the surface supra-basal layer. Thus, we found a unique K15 expression pattern in chicken reticulate scales: the basal layer cells expressed the stem cell marker in an unrestricted pattern not seen in other scale types. 

### 3.5. LRCs in Reticulate Scales Are Not Distributed in Clusters like in Avian Scutate Scales and Alligator Overlapping Scales 

We examined whether there are stem cell niches in reticulate scales. If so, are the properties of the stem cell niche different from the other scale types? To answer this question, we used 3 h BrdU pulse labeling to find transit-amplifying cells (TA cells) and the BrdU label retention method to locate slow-cycling cells, which are putative stem cells (Figure 5A,B). 

We first sought to identify the configuration of stem cells and TA cells in adult chicken reticulate scales by comparing their surface and hinge regions (Figure 5C). After 3 h pulse labeling, BrdU-positive cells were randomly distributed in the epidermis of both the hinge and surface regions (Figure 3D, red arrows). After 1-week BrdU pulse labeling, most basal keratinocytes (95% in the hinge region and 70% in the surface epithelium) were BrdU-positive (Figure 5E). After a 2-week chase period, none of the LRCs were detected in the basal layer of the epidermis in both the hinge and surface regions of the reticulate scales (Figure 5F). The 8-week chase period did not change this result (Figure 5G). 

In alligator tuberculate scales (Figure 5H), short-term BrdU labeling for 3 h detected proliferating cells that were randomly distributed in the epidermis of both the hinge and surface (red arrows in Figure 5I). After BrdU labeling for 1 week, 93% of basal layer cells in the hinge region were BrdU-positive, but only 13% were positive in the outer surface region (Figure 5J). These data suggest that the hinge epidermis has more cell proliferation than the surface region. After an 8-week chase period, there were LRCs in both the hinge and outer surface (Figure 5K). However, after a 16-week chase, the LRCs were only detected within the hinge region (Figure 5L, blue arrows). These results suggest that these two radially symmetric-shaped scales display different stem cell properties. LRCs do not exist in the mature chicken reticulate scales. Alligator tuberculate scales have similar localized LRCs in the hinge region to those in the overlapping scales. 

### 3.6. Wound Healing Response of Avian Foot Sole Skin and Reticulate Scales

To examine the regenerative ability of reticulate scales upon wounding, we surgically removed a full-thickness piece of footpad skin from an adult chicken. After 5 months, the wound site was covered with newly formed skin but without clear reticulate scale units (Figure 6A,A’). H&E staining showed that the wound region formed a flattened skin without the hinge or surface structures (Figure 6B) compared with the normal reticulate scale hinge (black arrows) and surface (white arrows). The Tenascin-C (*TNC*) expression level in the normal reticulate scales was higher in the surface dermis (green arrows) than in the hinge dermis (red arrows) (Figure 6C, normal part). However, the regenerated skin did not show this differential Tenascin-C expression pattern (Figure 6C, wound part). This result showed that the avian foot sole skin can repair and heal, but reticulate scale appendages do not regenerate. 

## 4. Discussion

### 4.1. The Molecular Expression in Embryonic Stages Specifies Different Adult Properties in Avian Scales

Our research indicates that various molecules expressed during embryogenesis are essential for the proper morphogenesis of scales that function during adulthood. We find that the molecular expression involved in the formative processes takes place during the early (E11) and middle (E14) stages of embryonic scale development. During scutate scale development, LGR6 at the early (E11) and middle (E14) stages is expressed exclusively in the surface region. In contrast, SOSTDC1 is expressed only in the hinge region. In reticulate scales, the expression pattern differences between the surface and hinge regions are not obvious (Figure 2). Therefore, we believe that reticulate scales do not have clearly demarcated boundaries between the surface and hinge. These findings suggest that these molecules play an important role in the morphogenesis of scutate but not reticulate scales during early embryonic stages. They may also play a crucial role in determining the configuration of stem cell niches in adults.

### 4.2. The Stem Cell Niche Configuration Determines the Mode of Physiological Regeneration and Response to Wounding 

Both mammalian hairs and avian feathers have a robust ability to regenerate through normal cycling. The follicular structure with a localized stem cell niche provides the potential for regeneration in the normal cycling and upon plucking [4,29]. Label retaining cells (LRCs) were found in the hinge of avian scutate and reptilian overlapping scales [8]. However, unlike hair and feathers, which have follicular structures with localized stem cells within their specific niche, these putative scale stem cells are diffusely distributed in the hinge. These overlapping-shaped scales have limited regenerative abilities. They can regenerate surface and hinge-like regions but cannot generate a real overlapping region [8]. Similar phenomena were observed in the overlapping scales in lizards [30,31,32]. The diffuse putative stem cells in the hinge may be responsible for the observed limited regeneration abilities in these overlapping scales. 

In reticulate scales, we could not detect the overlapping-scale-type diffuse putative stem cells in the hinge. This could explain the low regenerative ability of reticulate scales. Upon wounding, they may need clustered stem cells to reform the skin appendage units. 

### 4.3. Hinge versus Surface in Different Scale Architectures

In this study, we found that dome-shaped chicken reticulate scales and alligator tuberculate scales display different epidermal stem cell profiles. The molecular and cellular characteristics of alligator tuberculate scales are not yet well studied. We used tuberculate scales as a morphological control that may form using different molecular and cellular architecture. Our results show that alligator tuberculate scales have a similar LRC profile to alligator overlapping scales, which retain their putative stem cells in the hinge region. In contrast, chicken reticulate scales display a new profile that lacks LRCs. 

The hinge regions of avian and reptilian scales express alpha-keratins whereas the surface regions express beta-keratins [11,31,32]. The diffuse localized LRCs in the hinge are coupled with the K15 expression [8]. In the case of reticulate scales, basal layer cells in both the hinge and surface regions are K15-positive in a sporadic pattern, which implies that only a few reticulate scale basal layer cells are stem-cell-like. The K15 sporadic pattern in basal layer cells is coupled with the absence of LRCs, which suggests that the reticulate scales are a modified skin appendage with a special function (Figure 7). 

### 4.4. The Unique Feature of Reticulate Scales on Avian Foot Sole

The avian foot integument has to bear a large mechanical load and also sustains wear and tear through constant environmental friction interactions. It is important that they have functional forms that can bear these burdens. The inward epidermal ridge structures are only found in reticulate scales, not in scutate scales in chickens or overlapping/tuberculate scales in alligators. Reticulate scales are located on the ventral foot and toe pad, which is the tactile skin in chickens. Similar epidermal ridge structures can also be found in other mammals’ tactile skin, such as the human palm/plantar skin or mouse hind paw skin [33,34]. We speculate that these structures may provide mechanical resistance to vertical pressure, enabling birds to walk. It may also enlarge the cell proliferative capacity by increasing basal cell numbers in undulating structures. The evenly non-localized LRCs in reticulate scales may reflect the need for an effective way to replenish cells with a high turnover rate for wound healing but not enough time for regenerating reticulate scale appendages.

## 5. Conclusions

This study of the molecular and cellular characteristics of avian reticulate scales shows that they are quite different from avian scutate scales and alligator scales (Figure 7). The epithelium folding provides flexibility to bear mechanical forces. However, reticulate scales do not exhibit clear differences between their hinge and surface regions. The lack of clustered epidermal slow-cycling cells may be related to their low regeneration ability. They are more like complex epidermis, with different levels of epidermal folding to bear mechanical forces. They are less like skin appendages such as feathers, hairs, or scutate scales that have clustered stem cells and niches for regeneration. 

## Figures and Tables

**Figure 2 jdb-11-00030-f002:**
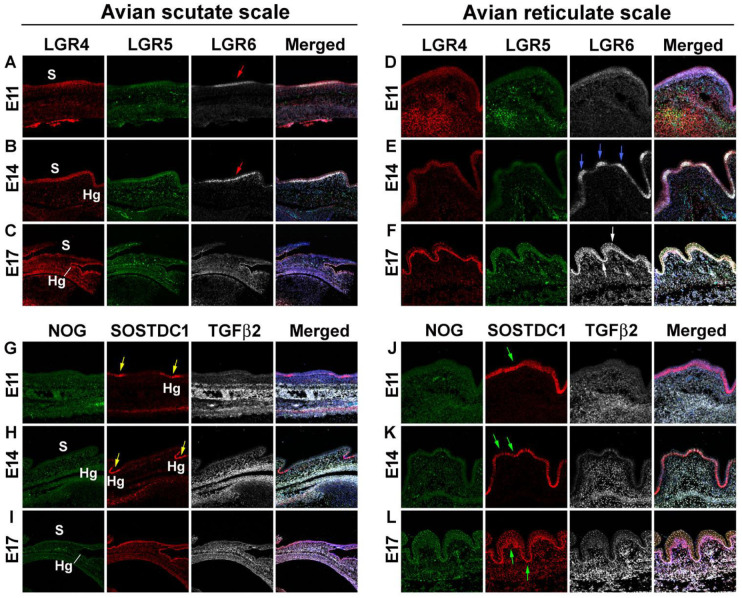
RNAscope analysis revealing differences in morphogen expression between avian scutate scales and reticulate scales. (**A**–**L**) RNAscope assay was used to detect specified RNA transcripts at embryonic day 11, 14, and 17. (**A**–**F**) LGR4, LGR5, and LGR6 staining in avian scutate (**A**–**C**) and reticulate scales (**D**–**F**). The fourth column shows the merged image including DAPI staining. Red arrows in A and B indicate LGR6 expression in the surface of scutate scales. Blue arrows in E indicate LGR6 expression in the surface of reticulate scales. White arrows in (**F**) show the wide LGR6 expression in reticulate scale epidermis. (**G**–**L**) NOG, SOSTDC1, and TGFβ2 staining in avian scutate (**G**–**I**) and reticulate scales (**J**–**L**). The fourth column shows the merged image including DAPI staining. Yellow arrows in G and H indicate SOSTDC1 expression in the scutate scale hinge area. Green arrows in (**J**–**L**) show the wide SOSTDC1 distribution in reticulate scale epidermis. Hg; hinge; S, surface.

**Figure 3 jdb-11-00030-f003:**
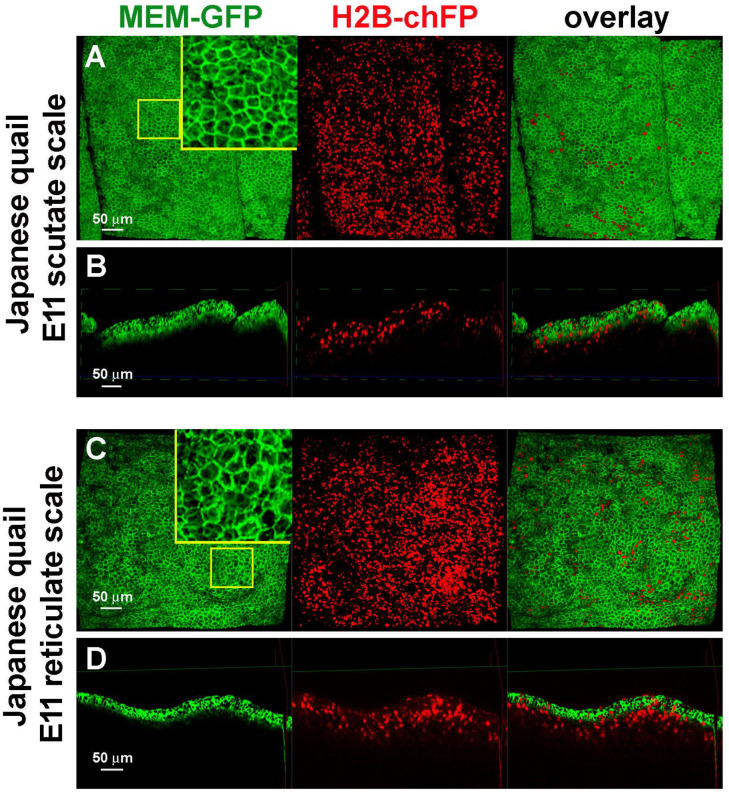
Confocal 3D images showing the development of scutate scale and reticulate scales using transgenic Japanese quail embryos. (**A**) Top view of scutate scale primordia at E11. Green expression of membrane-bound GFP and red expression of histone-bound Cherry. (**B**) Virtual transverse section view of scutate scale primordia at E11. (**C**) Top view of reticulate scale primordia at E11. (**D**) Virtual transverse section view of reticulate scale primordia at E11.

**Figure 4 jdb-11-00030-f004:**
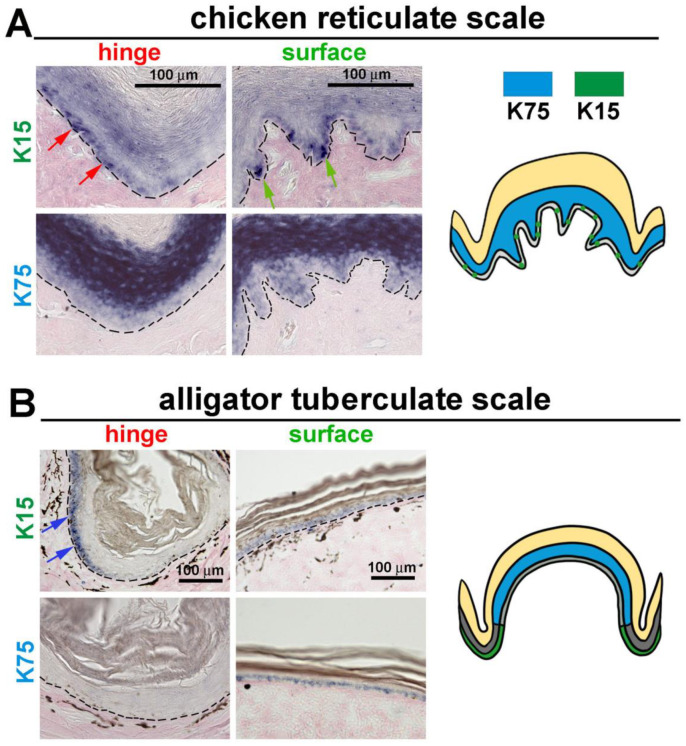
In situ hybridization of the stem cell marker K15 and differentiation marker K75 in avian reticulate and reptilian tuberculate scales. (**A**) K15 and K75 mRNA expression in chicken reticulate scales. Note: K15 is sporadically expressed in the basal epidermis of both the hinge (red arrows) and surface (green arrows) of chicken reticulate scales. (**B**) K15 and K75 mRNA expression in alligator tuberculate scales. Note: K15 is expressed in the hinge (blue arrows) but not in the surface regions of alligator tuberculate scales.

**Figure 5 jdb-11-00030-f005:**
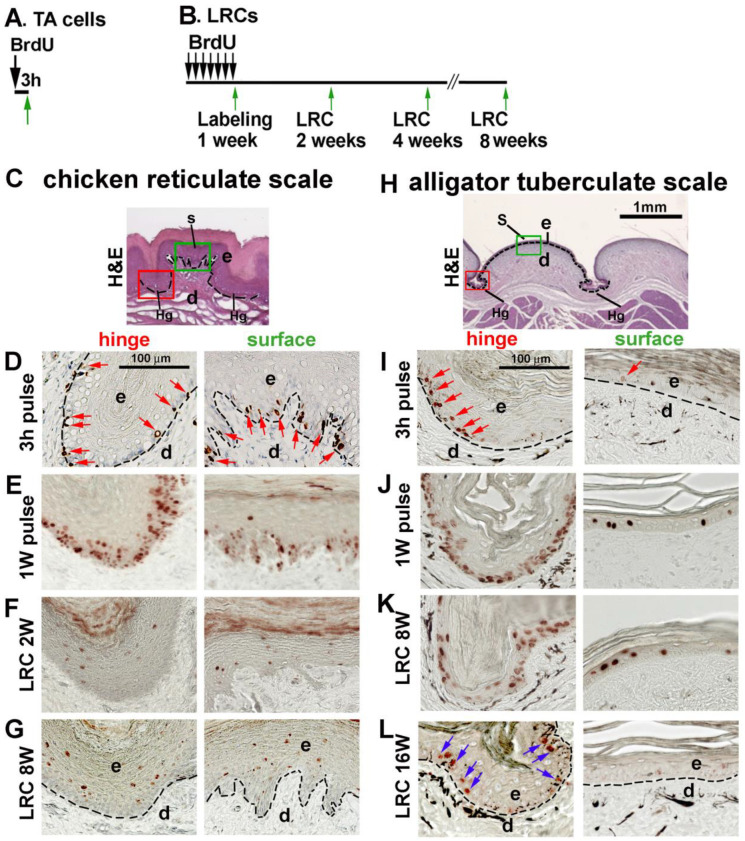
Topological distribution of putative stem cells in adult chicken reticulate scales and juvenile alligator tuberculate scales. (**A**) Strategy of TA cell labeling in adult chicken reticulate scales or juvenile alligator tuberculate scales. (**B**) Strategy of LRC labeling in adult chicken reticulate scales or juvenile alligator tuberculate scales. (**C**) H&E staining of adult chicken reticulate scales showing regions of interest. (**H**) H&E staining of juvenile alligator tuberculate scales showing regions of interest. The red and green rectangular boxes in panels (**C**,**H**) indicate the hinge region and surface region. (**D**–**G**) TA and LRCs in chicken reticulate scales. (**D**) 3 h pulse labeling. Red arrows indicate BrdU-positive cells. (**E**) BrdU 1-week labeling. (**F**) A 2-week chase period after 1 week of labeling. (**G**) An 8-week chase period after 1 week of labeling. (**I**–**L**) TA and LRCs in alligator tuberculate scales. (**I**) The 3 h BrdU pulse labeling. Red arrows indicate BrdU-positive cells. (**J**) BrdU labeling for 1 week. (**K**) An 8-week chase period after 1 week of labeling. (**L**) A 16-week chase period after 1 week of labeling. Blue arrows indicate the LRCs in the hinge region. Note that LRCs exist in the alligator tuberculate scale hinge regions but are negative in both the surface and hinge regions of chicken reticulate scales. d, dermis; e, epidermis; Hg, hinge; S, surface.

**Figure 6 jdb-11-00030-f006:**
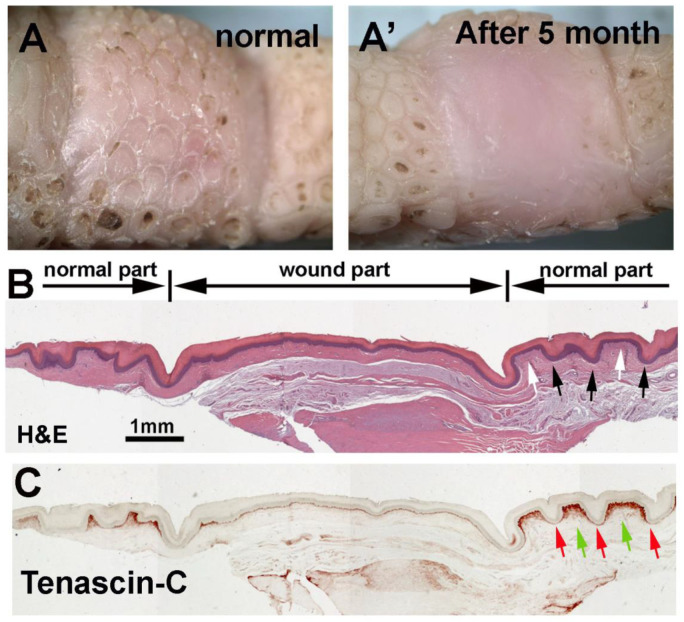
After full-thickness wounding, avian foot sole repairs the injury but does not regenerate reticulate scales. (**A**) Bright view of footpad reticulate scales before wounding and (**A**’) 5 months after wound healing. (**B**) H&E staining shows the structural differences between normal reticulate scales and the lack of scales within the wound healing region. (**C**) Tenascin-C immunostaining shows the wound region lacks differential dermal Tenascin-C expression between the normal hinge and surface. Black and white arrows indicate the normal reticulate scale hinge and surface, respectively. Red and green arrows indicate the reticulate scale dermis Tenascin-C expression in the hinge and surface regions.

**Figure 7 jdb-11-00030-f007:**
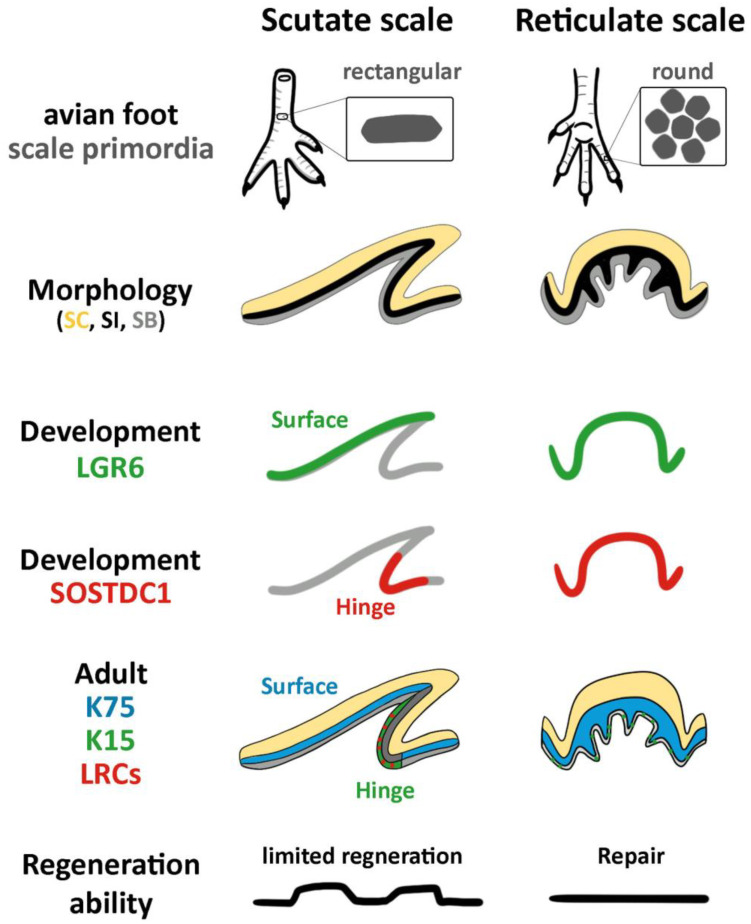
Summary of differences between scutate and reticulate scales in morphology, molecular expression, and regeneration ability. SB, stratum basal; SC, stratum corneum; SI, stratum intermedium.

## Data Availability

Original data are available upon request.

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
