# Peer review of "Molecular and Cellular Characterization of Avian Reticulate Scales Implies the Evo–Devo Novelty of Skin Appendages in Foot Sole"

_jdb, 2023, doi:10.3390/jdb11030030_

Round 1

Reviewer 1 Report

Dear authors,

thank you for the very interesting and useful for specialists manuscript which makes a new significant contribution to the study of skin derivatives of amniotes.This new scientific work confirm your high status as specialists in Evo-Devo. I clearly understand how much work it took to obtain this new data.

It will certainly decorate the special issue of the JDB dedicated to the anniversary of the professor L. Alibardi.

I sincerely wish you a successful publication of your manuscript and further success.

I noticed only  few inaccuracies in the text:

lines 112 and 302  - remove the dot at the end of the title

line 117 - It is desirable to specify the thickness of histological sections

Author Response

Reviewer 1:
Dear authors,
thank you for the very interesting and useful for specialists manuscript which makes a new significant contribution to the study of skin derivatives of amniotes.This new scientific work confirm your high status as specialists in Evo-Devo. I clearly understand how much work it took to obtain this new data.
It will certainly decorate the special issue of the JDB dedicated to the anniversary of the professor L. Alibardi.
I sincerely wish you a successful publication of your manuscript and further success.
I noticed only  few inaccuracies in the text:
lines 112 and 302  - remove the dot at the end of the title

Revised.

line 117 - It is desirable to specify the thickness of histological sections

Revised. In section 2.3, we now state: “For immunostaining, section in situ hybridization and RNAscope experiments, 7 µm paraffin sections were prepared.”

Reviewer 2 Report

From the title and abstract: this work focuses on the molecular and cellular characteristics of avian reticulate scales, to check whether the LRCs are distributed in specialized niches as feathers, hairs and avian scutate scales and reptilian overlapping scales. 

From the texts (e.g. results), the authors provide a intact data from the avian reticulate scale; but not always compared with the avian scutate scales and reptilian overlapping scales: e.g. 3.1, the avian reticulate scale compared with the avian scutate scale and alligator tuberculate scale; 3.2 and 3.3, compared with the avian scutate scale; 3.4 and 3.5 compared with the scutate scales and alligator overlapping scales from the published data,and the alligator tuberculate scale; 3.6 the results from wound healing response of avian foot sole skin and reticulate scales. 

Suggestions: 1. for 3.1, 3.2, 3.3, better provide a data set from the alligator overlapping scales. 

2. As long as the molecular and cellular characteristics of the alligator tuberculate scale is not well studied, it is not a good example to be used as a control. 

Author Response

Question 1: for 3.1, 3.2, 3.3, better provide a data set from the alligator overlapping scales.

We added alligator overlapping scale data in Fig. 1B (third column) and in section 3.1.
For 3.2, The RNAscope probes are designed for chicken samples and they do not work with alligator samples. We will examine alligator samples in future after we design and order alligator specific RNAscope probes.
For 3.3, we do not have transgenic alligator or fluorescence labeled alligator samples. We will think about an approach to address this question in the future.

Question 2:  As long as the molecular and cellular characteristics of the alligator tuberculate scale is not well studied, it is not a good example to be used as a control.

We agree that the molecular and cellular characteristics of the alligator tuberculate scale is not well studied. Since the alligator tuberculate scale has a similar dome shaped structure as chicken reticulate scales, we think it could be used as a morphological control that possibly forms using different molecular and cellular architecture. We pointed out this limitation in the discussion. In the future, we can pursue these characteristics in alligator tuberculate scales.

Reviewer 3 Report

The paper titled: “ Molecular and cellular characterization of avian reticulate scales implies the evo-devo novelty of skin appendages in foot soledescribes interesting differences in the morphology of scales depending on their location, and determines their ability to regenerate and rebuild. 

The manuscript is well organized and all figures and schematic drawings are at a good level.

Here are some comments.

Introduction:

The authors should consider possible research hypotheses to which they will clearly define the research objectives.

Page 2, line 73: The sentence is more of a conclusion and not a research objective.

Results:

Page 5 - the authors studied the expression of LGR4, 5, and 6, but only LGR4 and 6 are described. What results were obtained for LGR 5 in both scutate and reticulate scales? Do you see any difference expression of LGR 4,- 5, and 6 in the late developmental stage?

Page 5, lines 166-167; Page 6, lines 186-188; Page 8, lines 216-218; Page 9, lines 245-248; Page 11, lines 274-275 - all sentences are conclusions and should not be included in the section describing the results. The conclusions are interesting and authors should consider adding them in the discussion or in the conclusion of the paper.

Page 9, lines 221-228; page 11, lines 265-267 - these parts are a repetition of materials and methods, there is no need to write about them again.

Discussion:

It is quite short in relation to the interesting results obtained. The authors described the distribution of TA cells labeled by short-term BrdU during scale development and determined the expression of morphogens at early, middle, and late developmental stage. How do these results relate to the process of scale development? And how do they relate to the result obtained in an adult individual? The authors should consider further reflection and discussion of the obtained results.

Author Response

Reviewer 3:
The paper titled: “ Molecular and cellular characterization of avian reticulate scales implies the evo-devo novelty of skin appendages in foot sole” describes interesting differences in the morphology of scales depending on their location, and determines their ability to regenerate and rebuild.
The manuscript is well organized and all figures and schematic drawings are at a good level.
Here are some comments.
Introduction:
The authors should consider possible research hypotheses to which they will clearly define the research objectives.

We added: “We hypothesize that specific molecular and cellular characteristics of reticulate scales in foot pad, formed in the developing skin, set up the bio-architectural basis for adult birds to bear their weight bearing and to withstand constant friction.”

Page 2, line 73: The sentence is more of a conclusion and not a research objective.

We have rewritten the objective as follows: “In this paper, our objective is to assess the molecular expression and putative stem cell configurations of avian reticulate scales in comparison to other chicken and alligator scale types. We find that reticulate scales have different LRC properties and regeneration abilities compared to these other scale types.”  

Results:
Page 5 - the authors studied the expression of LGR4, 5, and 6, but only LGR4 and 6 are described. What results were obtained for LGR 5 in both scutate and reticulate scales? Do you see any difference expression of LGR 4,- 5, and 6 in the late developmental stage?

In section 3.2, we added this sentence, “In contrast, LGR5 did not show a specific expression pattern in developing scutate nor reticulate scales.”
In late developmental stage scutate scales, the expression of LGR4 in hinge is retained, but LGR6 is not expressed in both surface and hinge (Fig 2C). However, LGR4 is positive in both the surface and hinge in late stage reticulate scales (Fig 2F). LGR6 is expressed on the surface of reticulate scales in the middle stage (Fig. 2E, blue arrow) and then expands to the entire reticulate scale epidermis in the late stage (Fig. 2F, white arrow).

Page 5, lines 166-167; Page 6, lines 186-188; Page 8, lines 216-218; Page 9, lines 245-248; Page 11, lines 274-275 - all sentences are conclusions and should not be included in the section describing the results. The conclusions are interesting and authors should consider adding them in the discussion or in the conclusion of the paper.

We feel that a brief summary statement is helpful to the readers and would like to keep these sentences so the readers can more readily follow our logic.

Page 9, lines 221-228; page 11, lines 265-267 - these parts are a repetition of materials and methods, there is no need to write about them again.

We deleted the paragraph (line 237-240) describing the method to avoid repetition.
For page 11, lines 281-282, we rewrote this sentence to: “To examine the regenerative ability of reticulate scale upon wounding, we surgically removed a full-thickness piece of footpad skin from an adult chicken.”

Discussion:
It is quite short in relation to the interesting results obtained. The authors described the distribution of TA cells labeled by short-term BrdU during scale development and determined the expression of morphogens at early, middle, and late developmental stage. How do these results relate to the process of scale development? And how do they relate to the result obtained in an adult individual? The authors should consider further reflection and discussion of the obtained results.

We added a paragraph in discussion:
4.1 The molecular expression in embryonic stages specifies different adult properties in avian scales
Our research indicates that various molecules expressed during embryogenesis are essential for the proper morphogenesis of scales that function during adulthood. We find that molecular expression involved in the formative processes takes place during the early (E11) and middle (E14) stages of embryonic scale development. During scutate scale development LGR6 at early (E11) and middle (E14) stages are expressed exclusively in the surface region. In contrast SOSTDC1 is expressed only in the hinge region. Whereas, in reticulate scales, their expression pattern differences between the surface and hinge regions are not obvious (Fig. 2). Therefore, we believe that reticulate scales do not have clearly demarcated boundaries between the surface and hinge. These findings suggest that these molecules play an important role in the morphogenesis of scutate but not reticulate scales during early embryonic stages. They may also play a crucial role in determining the configuration of stem cell niches in adults.